# Genome-Wide Identification, Expression Analysis under Abiotic Stress and Co-Expression Analysis of *MATE* Gene Family in *Torreya grandis*

**DOI:** 10.3390/ijms25073859

**Published:** 2024-03-29

**Authors:** Hang Shen, Ying Hou, Xiaorong Wang, Yaru Li, Jiasheng Wu, Heqiang Lou

**Affiliations:** State Key Laboratory of Subtropical Silviculture, Zhejiang A&F University, Hangzhou 311300, China; 2021102091021@stu.zafu.edu.cn (H.S.); hymonkey@stu.zafu.edu.cn (Y.H.); 2021602122078@stu.zafu.edu.cn (X.W.); 2022102092010@stu.zafu.edu.cn (Y.L.)

**Keywords:** *Torreya grandis*, expression profiles, WGCNA analysis, MATE transporters

## Abstract

The multidrug and toxin efflux (MATE) family participates in numerous biological processes and plays important roles in abiotic stress responses. However, information about the *MATE* family genes in *Torreya grandis* remains unclear. In this study, our genome-wide investigation identified ninety *MATE* genes in *Torreya grandis*, which were divided into five evolutionary clades. *TgMATE* family members are located on eleven chromosomes, and a total of thirty *TgMATEs* exist in tandem duplication. The promoter analysis showed that most *TgMATEs* contain the cis-regulatory elements associated with stress and hormonal responses. In addition, we discovered that most *TgMATE* genes responded to abiotic stresses (aluminum, drought, high temperatures, and low temperatures). Weighted correlation network analysis showed that 147 candidate transcription factor genes regulated the expression of 14 *TgMATE* genes, and it was verified through a double-luciferase assay. Overall, our findings offer valuable information for the characterization of the *TgMATE* gene mechanism in responding to abiotic stress and exhibit promising prospects for the stress tolerance breeding of *Torreya grandis*.

## 1. Introduction

*Torreya grandis* is an evergreen conifer species of the Taxaceae family. It is an endangered plant species and only grows in a few mountainous areas of some southern provinces of China [1]. As the native economic nut tree species, the fruit of *Torreya grandis* has been documented to have many medicinal properties and has been used as food for thousands of years in China [2]. *Torreya grandis* seed oil is abundant in unsaturated fatty acids and bioactive compounds, rendering it of excellent nutritional and medical value [3]. Due to the high-value components, *Torreya grandis* seed oil has been widely employed in medicine, food, and cosmetics in Japan and China. [4,5]. Nut crops are an integral component of food production systems, which are vital to nutrition and food security [6]. However, the frequent occurrence of abiotic stress environments in recent years has shown adverse effects on crop yields and thus poses a challenge to food security [7,8]. In recent years, acid rain and long-term improper fertilization have resulted in increased soil acidification and aluminum (Al) toxicity stress in *Torreya grandis* plantations, which adversely affects the yield and quality of *Torreya grandis* [9,10]. In addition, severe drought occurrences are common in *Torreya grandis* planting areas from the summer to autumn, causing *Torreya grandis* seedling growth and development to be hampered by water stress, which inhibits their growth and development [2]. Therefore, it is important to improve the resistance of *Torreya grandis* to various abiotic stresses. Nowadays, genetic engineering is widely used for crop improvement and provides a means to address the challenges of food security associated with environmental stress [11]. Numerous studies have shown that molecular breeding methods, such as genetic modification, can enhance plant stress tolerance traits [12,13]. Consequently, the breeding of *Torreya grandis* with greater abiotic stress tolerance is important for ensuring food security.

Over a long period of evolution, plants have evolved plenty of strategies to adapt to abiotic stress. When subjected to external stress, the stress-specific signal transduction pathways in plants will be triggered, thus activating relevant regulatory mechanisms to adapt themselves to the stress [14]. One of the regulatory mechanisms is the differential expression of stress-related genes [15]. The molecular processes of plant responses to abiotic challenges have been widely explored, and many genes critical for improving tolerance to various abiotic pressures have been found. The *MATE* gene family, which encodes multidrug and toxic compound extrusion transporters, is believed to be relevant to numerous abiotic stresses [16].

The multidrug and toxic compound extrusion (MATE) family is a newly classified family of multidrug efflux transporters that are ubiquitous in both prokaryotes and eukaryotes [16,17]. In plants, MATE transporters are engaged in a variety of biological processes [16]. They were discovered to have an essential role in the transport of metabolites, including flavonoids and alkaloids [17]. For example, *Arabidopsis TT12* encoding the MATE protein was found to be able to transport flavonoids, which are involved in the vacuolar accumulation of proanthocyanidin precursors in the seed [18]. In addition, MATE transporters have been proven to be involved in the endogenous and exogenous mechanisms of detoxification [19] and the transport of plant hormones, such as efflux abscisic acid (ABA) [20] and salicylic acid (SA) [21]. At present, *MATE* family members have been discovered in many angiosperms, such as *Oryza sativa* [22], *Solanum tuberosum* [23], and *Arabidopsis thaliana* [24]. Studies in these species have revealed that plant *MATE* genes play a key role in the response to various abiotic stresses. In *Arabidopsis thaliana*, *MATE* family member *AtDTX50* can promote cellular ABA efflux and consequently impact drought tolerance [20]. The barley *MATE* gene, *HvAACT1*, could accelerate the citrate efflux and improve the Al tolerance of the transgene plants [25]. However, the MATE family has not been discovered in gymnosperms to date. The structural features of *MATE* genes in *Torreya grandis* and their effects in the response to abiotic stress need to be further studied. Thankfully, the whole genome sequencing of *Torreya grandis* has been completed recently [26], which provides the basis for analyzing the MATE family in gymnosperms.

In this investigation, we aimed to conduct a comprehensive genome-wide analysis of the *Torreya grandis MATE* gene family. We systematically analyzed their phylogenetic relationships, gene structure, chromosome localization, potential cis-acting elements, and expression profiles in Al, drought, high-temperature, and low-temperature responses. In addition, we investigated potential transcription factors that regulate *TgMATE* members’ expression under abiotic stress through weighted correlation network analysis (WGCNA) and validated their reliability through a dual-luciferase assay. In summary, the response of *TgMATE* genes to abiotic stress was initially investigated in this research, which is expected to be helpful for improving the stress tolerance of *Torreya grandis* through genetic engineering.

## 2. Results

### 2.1. Identification and Characterization of MATE Genes in Torreya grandis

In our research, there are 90 *MATE* family members that were identified in *Torreya grandis* through HMM search and conserved domain analysis. We named those genes *TgMATE1*-*TgMATE90* according to their chromosomal locations. To comprehend the physicochemical characteristics of TgMATE proteins, a bioinformatics evaluation was carried out. The outcomes indicated that TgMATE proteins have an amino acid count ranging from 75 to 755 aa. The molecular weight of these proteins varies from 8.253 to 83.36 kDa, while the isoelectric point (pI) ranges from 4.74 to 10.42. According to the subcellular localization prediction results, 69 TgMATE proteins are located on the plasma membrane, while less than a quarter of the *TgMATEs* are distributed on other organelles (Appendix A).

### 2.2. Chromosomal Localization of TgMATEs

TBtools was used to process the genome annotation file and visualize the chromosome distribution of *TgMATE* genes [27]. The results showed that the *TgMATEs* gene was distributed on all 11 chromosomes of *Torreya grandis*. Chromosome 5 contains the largest number of *TgMATE* genes, which has 19 *TgMATEs*, while the least number of *TgMATE* genes distributed is chromosome 9, which has just 1 *TgMATE* gene (Figure 1). The processes of gene replication are crucial to the evolution of organisms [28]. Tandem duplication, one of the gene replication processes, is defined by the coexistence of family members in the same or nearby intergenic areas, and it might encourage the growth of gene families [29]. The tandem duplicates in the *TgMATE* gene family were investigated through the MCScanX tool [30], and those genes were marked with red arcs (Figure 1). Finally, there are a total of 30 tandem repeat *TgMATE* genes that were found, which constituted 20 tandem duplication events. The outcomes indicate that there were numerous tandem duplication events that occurred, which fueled the growth of the *TgMATE* genes in *Torreya grandis*.

### 2.3. Phylogenetic Analysis of TgMATE Proteins

A phylogenetic tree was constructed based on the protein sequences of 18 different species to evaluate the evolutionary relationship of the *TgMATE* gene family. These MATE proteins were categorized into five clades (I–V), which are illustrated in Figure 2. The *TgMATE* genes were distributed in all clades, and clade Ⅴ has the largest number of *TgMATEs* (49 *TgMATEs*). In clades I, Ⅱ, and Ⅲ, there are 24, 6, and 4 *TgMATE* genes, respectively. Clade Ⅳ (seven *TgMATEs*) contains the most abundant species, with a total of fourteen plant species of MATE proteins, indicating that they may have important functions and are relatively conservative in evolution. Interestingly, we found that many homologous MATE genes in clade Ⅳ have been reported to be associated with tolerance to Al stress, including *TaMATE1B* [31], *SbMATE* [32], *VuMATE1* [33], *VuMATE2* [34], *OsFRDL4* [35], *HvAACT1* [36], *PtrMATE1,* and *PtrMATE2* [37]. These results imply that the *TgMATE* genes in clade Ⅳ may also have resistance to Al stress.

### 2.4. Investigation of Protein Motifs and Gene Structure of TgMATEs

We created the exon–intron structure diagram of *TgMATE* genes according to the *Torreya grandis* genome annotation file to investigate the structural variety of the *TgMATE* family members (Figure 3B). The exon number of the *TgMATE* gene ranges from 2 to 14. The bulk of the *TgMATE* genes in clades II, Ⅲ, and IV have quite lengthy intron sequences, whereas the majority of *TgMATE* genes in clades Ⅰ and V have practically short intron sequences. Noticeably, the *TgMATE* genes with close genetic distances have comparable exon–intron structures, whereas the structural patterns of *TgMATE* genes on different phylogenetic branches are distinct. Additionally, the evolutionary characteristics of TgMATE proteins were further explored through the conserved domain analysis in MEME, and 10 conserved motifs have been discovered (Figure 3C). Figure 3A shows the distribution of conserved motifs on TgMATE proteins. TgMATE proteins with nearly genetic distances have similar conserved motif categories and distributions. The protein motif composition of clades III and IV differed dramatically from that of clades I, II, and V. Members of clades III and IV, for example, have just 1–2 motifs in common, while TgMATE proteins in clades I, II, and V have 2–10 motifs in common, and most of them have 10 conserved motifs. Our findings are compatible with phylogenetic analysis, which supports the categorization of *MATE* family members.

### 2.5. Cis-Acting Elements Analysis of TgMATE Gene Promoters

After the prediction in PlantCARE, a total of 67 cis-acting elements with known functions were identified in the *TgMATE* gene promoters. According to their main functional characteristics, these elements were classified into six categories, encompassing the environmental stress-related, hormone-responsive, light-responsive, development-related, site-binding-related, and promoter-related elements (Appendix A and Figure 4). As the results demonstrate, the light-responsive elements were the most abundant cis-acting elements in *TgMATE* promoters, accounting for 31 different types, followed by the hormone-responsive elements, accounting for 11 cis-elements. There are seven categories of environmental stress-related elements that were detected, which were DRE, LTR, ARE, GC-motif, TC-rich repeats, MBS, and WUN-motif in the *TgMATE* promoters. The development-related elements include eight important types, such as cell cycle regulation, circadian control, etc. Notably, all the *TgMATE* promoters contain light-responsive cis-elements. In addition, the stress-response cis-elements exist in almost all *TgMATE* promoters, except for the promoters of *TgMATE44, TgMATE69, TgMATE70,* and *TgMATE87* (Appendix A). Moreover, 83 *TgMATE* promoters have hormone-responsive elements, and 59 *TgMATE* promoters have development-related elements (Appendix A).

### 2.6. Analysis of the Expression Profiles of TgMATEs under Various Abiotic Stresses

To obtain some insights into how *TgMATE* genes respond to abiotic stress, we analyzed the transcript profile of *TgMATEs* based on the transcriptome data under Al, drought, high-temperature, and low-temperature stress. A total of 63 *TgMATEs* were found to have differential expression (Fold Change > 1.5 or <0.8) in the four types of abiotic stress treatment (Appendix A). As shown in Figure 5A, the number of up-regulated and down-regulated *TgMATE* genes was on the same order of magnitude in Al and low-temperature stress (around 20 *TgMATEs*), while the quantity of down-regulated *TgMATE* genes is significantly greater than that of up-regulated *TgMATE* genes under drought and high-temperature stress. Next, we contrasted the overlapping relationships of the differentially expressed genes (DEGs) under different stress situations through the Venn diagram (Figure 5B). It is interesting to note that 18 *TgMATE* genes responded to Al, drought, high-temperature, and low-temperature stress, and 25 *TgMATE* genes responded to at least two stresses (Figure 5B), suggesting *TgMATE* family genes are extensively implicated in abiotic stress response processes in *Torreya grandis*.

Further, we constructed the gene expression heat maps to illustrate the expression profile of *TgMATE* genes under different stresses (Figure 5C–F). During Al stress, the patterns of these *TgMATE* genes’ expression can be divided into four groups (patterns Al_1–4) (Figure 5C). Expression pattern Al_1 exists in 12 *TgMATE* genes, where their expressions reached the highest levels at 8 h but decreased at 24 h after AlCl_3_ treatment (Figure 5C). Pattern Al_2 comprises 13 *TgMATEs* whose expression levels increased during the Al treatment and peaked at 24 h (Figure 5C). Pattern Al_3 contains nine *TgMATEs*, and their expression was lowest at 8 h (Figure 5C). Pattern Al_4 (14 *TgMATEs*) displayed the opposite expression pattern to pattern Al_2, with relatively high expression levels at 0 h of Al treatment but decreasing after 8 h and 24 h of Al treatment (Figure 5C). During drought stress, pattern Dr_1 (13 *TgMATEs*) exhibited comparatively high expression at 60 d compared to 0 d and 40 d (Figure 5D). Pattern Dr_2 (18 *TgMATEs*) was generally down-regulated at 40 d and 60 d (Figure 5D). Pattern Dr_3 (6 *TgMATEs*) was increased at 40 d but decreased at 60 d (Figure 5D). For high-temperature conditions, pattern HT_1 (17 *TgMATEs*) exhibited an up-regulation trend, while a majority of genes in pattern HT_2 showed significant down-regulation at 1 d, 2 d, and 4 d of high-temperature treatment (Figure 5E). Moreover, under low-temperature stress, pattern LT_1 consisting of 20 *TgMATEs,* was up-regulated at nearly all the time points (Figure 5F). Most genes in pattern LT_2 were down-regulated at 2 d, 4 d, and 6 d of low-temperature treatment, while in contrast, a few of them were up-regulated at 6 d of low-temperature treatment (Figure 5F). Pattern LT_3 (6 *TgMATEs*) positively responded to low-temperature stress in the early phases but was down-regulated during the later phases (Figure 5F). These findings highlight the possibility of these *TgMATEs* involved in Al, drought, high-temperature, and low-temperature stress responses and imply the different contributions of TgMATE family members to abiotic stress tolerance in *Torreya grandis*.

### 2.7. Expression Pattern of TgMATEs in Different Phylogenetic Clades under Abiotic Stresses

We also examined the expression patterns of each phylogenic clade of *TgMATE* family genes in the four types of abiotic stress. Clade I has a total of 24 *TgMATEs*, among which 13, 6, 9, and 9 genes respond in Al, drought, high-temperature, and low-temperature, with proportions of 54.17%, 25%, 37.5%, and 37.5%, respectively (Figure 6). There are two *TgMATEs* (33.33%) in clade II responding to Al, drought, and low-temperature stress, and five *TgMATEs* (83.33%) responding to high-temperature treatment (Figure 6). Clade III contains only four *TgMATEs*, with one gene (25.0%) responding in Al and low-temperature stress, three genes (75.0%) in drought stress, and all of them responding in high-temperature stress (Figure 6). Clade IV has more than half of *TgMATEs* that respond to each type of stress, with five (71.43%), four (57.14%), seven (100.0%), and seven (100.0%) genes in Al, drought, high-temperature, and low-temperature stress, respectively (Figure 6). Clade V contains the largest number of *TgMATE* members (49 in total), and 17, 20, 26, and 24 *TgMATEs* in this clade responded under the four treatments, accounting for 34.69%, 40.82%, 53.06%, and 48.98%, respectively (Figure 6). According to our results, different evolutionary branches of the TgMATE family exhibit varying proportions of response genes to abiotic stress, indicating that these branches may contribute differently to the abiotic stress response. Interestingly, clade IV has the highest proportion of *TgMATEs* responding in Al, high-temperature, and low-temperature treatments of the five polygenetic branches, implying that *TgMATE* members in this clade are widely involved in *Torreya grandis* resistance to abiotic stress. Additionally, clade Ⅲ showed the highest percentage of response genes to drought stress, suggesting that this clade may be more resistant to drought stress. Our findings demonstrate the broad involvement of the TgMATE family in the response to abiotic stress and imply that genes from various evolutionary branches may contribute differently to facing abiotic stress, which offers possible candidate genes for *Torreya grandis* stress-tolerant breeding.

### 2.8. QRT-PCR Validation

To validate the results of transcriptome analysis, we evaluated the expressions of the *TgMATE* family members in clade Ⅳ under Al treatment by qRT-PCR. Unsurprisingly, the expression patterns of these *TgMATEs* obtained from qRT-PCR almost matched the outcomes of the transcriptome study. During the Al stress treatment, the expressions of *TgMATE1*, *TgMATE2*, and *TgMATE4* dramatically rose, reaching their peak levels at 24 h (Figure 7). *TgMATE87* experienced a significant upregulation at 8 h while exhibiting a downregulation at 24 h (Figure 7). These results demonstrated the reliability of the expression pattern of *TgMATEs* under various abiotic stresses obtained through transcriptome data.

### 2.9. Identification of the Potential Transcription Factors by WGCNA

To investigate the reactions between *TgMATEs* and the transcript factors in *Torreya grandis*, a WGCNA was completed using the transcriptome data of *Torreya grandis*. A cluster tree was constructed according to the correlation between gene expression levels, and 11 different modules were generated by the dynamic cutting method (Figure 8A,B). Correlation weights calculated by WGCNA were used to measure the interaction degree between transcription factors and *TgMATE* genes. We chose a weight threshold of 0.01 as a criterion for identifying transcription factors that might interact with *TgMATEs* and used Cytoscape to visualize their co-expression relationship. According to the WGCNA analysis, a total of 147 transcription factors were found to have an association with 14 *TgMATE* family members (Figure 8C and Appendix A). The transcription factors were arranged into 28 families, and the majority of these transcription factors belong to WRKY, bHLH, HD-ZIP, bZIP, NAC, and ERF families (Appendix A), implying that these transcription factor families may play an important role in regulating *TgMATE* gene expression.

### 2.10. Verification of WGCNA Results through the Dual-Luciferase Reporter Assay

To verify the prediction results of WGCNA, *TgMATE2* and the transcription factor predicted to be associated with it in the turquoise module were selected for the dual-luciferase reporter assay. Through PCR reaction, we successfully amplified the *TgMATE2* promoter fragment and four transcription factors (*evm.TU.PTG000304L.6, evm.TU.PTG009519L.1, evm.TU.PTG000725L.65,* and *evm.TU.PTG016833L.4*) from *Torreya sinensis* DNA and cDNA. Their sequences were cloned into the reporters and the effectors, respectively, followed by the transient co-expressed in tobacco leaves through *Agrobacterium tumefaciens*. The empty 62-SK plasmid was used as the negative control. LUC and Rluc values were measured, and the interaction intensity of the *TgMATE2* gene promoter region and the transcription factor proteins was reflected by the ratio of LUC to Rluc. As shown in Figure 9, the transcription factors in the co-expression network had different activation capacities for *TgMATE2* promoters. Notably, transcript factor *evm.TU.PTG016833L.4* from the HD-ZIP family, had a significantly higher LUC/Rluc ratio than the negative control (Figure 9B). In addition, the fluorescence of *evm.TU.PTG016833L.4* detected through Tanon 5200 is brighter than that of the control (Figure 9C), implying this HDZIP transcription factor could activate the expression of *TgMATE2*. These findings support the validity of the approach for identifying transcription factors that control *TgMATE* members via WGCNA.

## 3. Discussion

### 3.1. Torreya Grandis Genome Possessed a Large TgMATE Gene Family

Our research discovered 90 *TgMATE* family members by searching the *Torreya grandis* genome with the hidden Markov model of the *MATE* protein. According to earlier studies, there are 56 *MATE* family genes in *Arabidopsis* [24], 46 in *Oryza sativa* [22], 42 in *Cucumis melo* [38], 48 in *Solanum tuberosum* [39], and 42 in *Capsicum annuum* [23]. Compared to these reported species, *Torreya grandis* has a substantially higher number of *MATE* family genes, suggesting their diversity and important role in plants. Gene replication is one of the main driving forces for the evolution and expansion of gene families [40]. Previous studies have shown that tandem and fragment replication mainly promote the replication of the *MATE* gene family [22,23,41]. The collinearity analysis of the *Torreya grandis* genome revealed there are 30 (33.4%) tandem repeat genes in the *TgMATEs*, which was close to the number of tandem repeat *MATE* genes in pepper and potato (38.1% and 35%, respectively) [23]. Therefore, we consider that tandem duplication may also promote the expansion of the *MATE* gene family in *Torreya grandis*. The results of subcellular localization prediction showed that the majority of *TgMATEs*, 69 in number, were located on the plasma membrane. This is compatible with the function of toxic compound efflux.

The length of MATE family proteins varies in different species. *Arabidopsis* MATE proteins contain 414 to 539 residues [24], and OsMATE proteins range in length from 370 to 598 aa [22]. MATE proteins in *Gossypium arboreum* range from 153 to 722 aa [42]. Our research discovered that the amino acid length of the TgMATE protein varied from 75 to 755, implying that the TgMATE family has higher diversity and complexity. According to phylogenetic analysis, MATE proteins from eighteen species could be divided into five clades, which is consistent with the *MATE* gene family previously reported in *Capsicum annuum* [23]. Further analysis reveals that the conserved motif distribution and gene structure of *TgMATEs* in different clusters were significantly distinct, while similar patterns were found among *TgMATE* genes in the same phylogenetic branch (Figure 3), which supports the classification results of MATE protein in the phylogenetic analysis. In summary, the differences among *TgMATE* family members suggest their diverse gene functions. Exon and intron play an important role in the differentiation of gene structure and function [42]. The divergence of gene structure could be primarily caused by the exonization of intronic sequences or pseudoexonization of exonic sequences [43]. We noticed *TgMATE* genes have very long intron lengths, especially in clades II, III, and IV. In tobacco, the intron length of *MATE* family genes is within 30 kb, and some members are intron-less [44]. While in *Torreya grandis*, all *TgMATE* genes have introns, and some are even more than 100 kb in length. This suggests that the expansion of *TgMATE* genes may be controlled by changes in the number and length of introns.

### 3.2. Members of the TgMATE Family Play an Essential Part to Response the ABIOTIC Stress in Torreya grandis

Nowadays, abiotic stressors, including Al, drought, high-temperature, and low-temperature stresses, are recognized as significant barriers to plant growth and agricultural productivity. Understanding the molecular response mechanism of plants under stress is helpful for the development of plant resistance breeding [45]. It is generally believed that one of the means for plants to cope with Al stress is to form complexes with extracellular aluminum ions through the secretion of citrate and other organic acid anions in the roots to alleviate Al toxicity stress [46]. Under drought stress, plants will accumulate a large amount of ABA and regulate the expression of downstream-related genes to resist drought [47]. In addition, plants will produce many reactive oxygen species under abiotic stress, resulting in tissue oxidative damage [48]. Numerous studies have shown that MATE transporters can respond to a variety of abiotic stresses through these mechanisms [15,31,32,49]. Nevertheless, little is known about how the *TgMATE* genes in *Torreya grandis* respond to abiotic stress. To date, transcriptome analysis of gene expression profiles has become one of the most important means of biological research and is generally considered to provide important clues for the exploration of gene functions [40]. Therefore, we investigated the expression pattern of *TgMATE* genes in response to Al, drought, high-temperature, and low-temperature stress based on the transcriptomic data. We found that more than half of *TgMATE* genes (70%) were differentially expressed under abiotic stress. The number of up-regulated genes in Al and low-temperature stress is almost equal to that of down-regulated genes, whereas the number of up-regulated genes is less than that of down-regulated genes in drought and high-temperature stress, suggesting that *TgMATE* genes tend to be down-regulated in response to drought and high-temperature stress in contrast to Al and low-temperature stress. Under Al stress, we found that 19 *TgMATEs* were up-regulated and 22 *TgMATEs* were down-regulated. There is some evidence showing that MATE transporters are responsible for the secretion of organic anion citrate, which affects the Al tolerance of plants. For example, the introgression of bread wheat *TaMATE1B* into durum wheat enables better growth in the soil with high Al^3+^ [31]. The transgenic sugarcanes with the *SbMATE* gene overexpressed have an improved tolerance to Al [32]. These data suggested that *TgMATE* family genes might be involved in Al stress response. The *Arabidopsis* homologous gene *DTX50* was found to encode a transporter with ABA effector function and thus play a significant role in response to drought stress [48]. In our research, 15 *TgMATEs* were up-regulated and 22 *TgMATEs* were down-regulated. Under drought treatment, suggesting the possibility of these *TgMATEs* responding to drought stress. high-temperature and low-temperature stress are common abiotic stresses that accumulate reactive oxygen species and cause damage to plants. Previous studies have shown that the Cotton *DTX/MATE* gene accelerates the expression of antioxidant genes, which reduces the content of reactive oxygen species and enhances abiotic stress tolerance in transgenic Arabidopsis [15]. In *Torreya grandis*, there are 11 and 24 genes that were up-regulated, and 44 and 22 genes were down-regulated under high-temperature and low-temperature stress, respectively, indicating their important roles in responding to extreme temperature. Notably, 25 *TgMATEs* responded to at least two stresses, and 18 *TgMATEs* responded to all four types of stress, indicating that *TgMATE* family members actively participate in the abiotic stress response mechanisms in *Torreya grandis*. Moreover, hierarchical clustering results showed that *TgMATEs* had different expression patterns under four different stresses, implying that the *TgMATE* genes may contribute differently to cope with these abiotic stresses. In summary, these findings suggest that the *TgMATE* genes are extensively associated with the abiotic stress response in *Torreya grandis* and could provide the molecular foundation for the breeding of *Torreya grandis* cultivars with improved abiotic stress tolerance.

### 3.3. Expression Profiles of Different TgMATE Subfamilies under Abiotic Stress

During evolution, different phylogenic clades within a gene family typically acquire distinct functions to enhance their adaptability to changing environmental conditions [50]. In Arabidopsis and rice, the subfamily MATE III has distinct physiochemical properties that are different from the other subfamilies, as Type-II divergence only occurred in the MATE III subfamily, while there were no Type-II functional divergence sites in any other subfamilies [51]. Nevertheless, little research has been conducted on how *MATE* family members from different phylogenic clades react to diverse abiotic stresses. Here, we analyzed the expression patterns of *TgMATE* family members during abiotic stress to explore their functional differentiation in abiotic stress responses. *TgMATE* family members from five evolutionary clades have different numbers of responses across the four stress types. For example, clade IV has the highest percentage of *TgMATE* genes responding in Al, high-temperature, and low-temperature stress, which suggests that *TgMATEs* in this clade are extensively implicated in plant resistance to abiotic stress. Notably, clade Ⅳ also combines many MATE homolog genes that have been confirmed in other plants to be tolerant to Al resistance, e.g., *VuMATE1* [33], etc., which highlights the important roles of *TgMATEs* in clade IV in the abiotic stress. Moreover, we found clade Ⅲ had the highest proportion of response genes under drought stress, indicating this clade may tend to resist drought stress. Our findings suggest that different TgMATE family evolution branches may contribute differentially to the abiotic stress tolerance in *Torreya grandis*.

### 3.4. Mining of Candidate Transcription Factors That Interact with TgMATEs

Transcription factors control gene expression at the transcriptional level by attaching to cis-acting elements on target promoters [52]. In plants, transcription factors are essential in numerous biological activities such as the regulation of development, the induction of defense, and the stress response [53]. Recent studies have reported that transcription factors regulate the expression of the *MATE* gene in other plants. In rice, Yokosho et al. [54] discovered that a c2h2 type zinc finger Al tolerance transcription factor *ART1* controls the expression of the rice *MATE* gene (*OsFRDL4*). However, the transcription factors that regulate the family members of *TgMATEs* in *Torreya grandis* have not yet been discovered. In this study, transcription factors co-expressed with *TgMATE* genes were predicted through WGCNA analysis. A total of 147 transcription factors were discovered to potentially regulate the expression of *TgMATE* genes and their relationship network was constructed. Many of these transcription factors belong to the WRKY, bHLH, HDZIP, bZIP, NAC, and ERF families, which were believed to be widely involved in plant responses to stress. For instance, wheat *TaWRKY10* could regulate the expression of a series of stress-related genes to improve the stress resistance of tobacco [55]. In *Vigna umbellate*, an NAC-type transcription *VuNAR1* could regulate the Al resistance through interaction with the cell wall-associated receptor gene promoter to regulate the metabolism of cell wall pectin [56]. The two HDZIP family members, *AtHB7* and *AtHB12*, oppositely control Al resistance by affecting the root cell wall Al accumulation in *Arabidopsis thaliana* [57]. Therefore, we speculated that the transcription factors screened by WGCNA analysis could respond to abiotic stress and regulate the expression of *TgMATE* genes, thus affecting the stress tolerance of *Torreya grandis*. Furthermore, the dual-luciferase reporter assay showed that a transcription factor (*evm.TU.PTG016833L.4*) from the HDZIP family in the co-expression network can positively regulate the expression of *TgMATE2*, suggesting the credibility in mining the transcription factors of *TgMATEs* through WGCNA. The potential transcription factors predicted by WGCNA can provide a molecular basis for further exploring the mechanism by which *TgMATE* genes respond to abiotic stress. Above all, the study of transcription factors related to *TgMATE* genes has important biological significance; it could not only enrich our knowledge of the mechanism of *Torreya grandis* in response to environmental stress but also contribute to providing new theoretical guidance for the genetic improvement of *Torreya grandis* in stress conditions.

## 4. Methods

### 4.1. Identification and Characterization of the MATE Family Members in Torreya grandis

The entire genome sequence of *Torreya grandis* and the annotation file were downloaded from Figshare (https://doi.org/10.6084/m9.figshare.21089869.v1, accessed on 29 April 2023) [58]. To find the potential *TgMATE* genes, we obtained the Hidden Markov model file of MATE protein (PF01554) in PFAM (https://pfam-legacy.xfam.org, accessed on 29 April 2023) and used the HMMER3.0 software to employ an HMM search (E-value < 1 × 10^−5^) on the *Torreya grandis* proteomes. The sequences of those putative TgMATE proteins were validated further based on their MATE domain, which was discovered through the NCBI web tool CDD-Search (https://www.ncbi.nlm.nih.gov/Structure/bwrpsb/bwrpsb.cgi, accessed on 29 April 2023). The isoelectric point (PI) and molecular weight (MW) were determined via the ExPASY online tools (https://web.expasy.org/protparam/, accessed on 8 May 2023) [59]. We used the TargetP 2.0 web tool (https://services.healthtech.dtu.dk/services/TargetP-2.0, accessed on 8 May 2023) [60] to forecast the protein subcellular location of each *TgMATE*.

### 4.2. Phylogenetic Analysis

Homologous MATE protein sequences in other plants were obtained from NCBI. The proteins of 18 species were used for the phylogenetic analysis, including 11 dicotyledonous plants and 6 monocotyledons in addition to *Torreya grandis*. We aligned all protein sequences through ClustalW [61], and the phylogenetic tree was constructed based on the Maximin Likelihood method with the bootstrap value 1000 in IQTREE software v1.6.12 [62]. The phylogenetic relationship of these MATE proteins was visualized through ITOL (https://www.itol.org/, accessed on 11 July 2023) [63].

### 4.3. Chromosomal Location, Gene Structure, and Conserved Motifs Analysis

We used TBtools software v2.042 to extract the exons, introns, and UTR information of *TgMATEs* from the *Torreya grandis* genome annotation file [27]. The 10 most conserved motifs of TgMATE proteins were detected through MEME SUITE (https://meme-suite.org/meme/, accessed on 13 May 2023) [64]. The chromosome distribution records of *TgMATEs* were derived via the genome annotation file. Finally, these outcomes were visualized via the TBtools software v2.042 [27].

### 4.4. Analysis of the Promoter Cis-Acting Elements of TgMATEs

Based on the genomic annotation file, the promoter sequence located in the upper 2000 bp of the coding sequence of the *TgMATEs* was extracted. The cis-acting elements of their promoters were predicted through PlantCARE websites (http://bioinformatics.psb.ugent.be/webtools/plantcare/html/, accessed on 9 May 2023) [65], and the results were visualized through TBtools software v2.042 [27].

### 4.5. Transcriptomic Analyses of TgMATE Genes

Our lab has previously constructed a transcriptome database of 1-year-old *Torreya grandis* at various abiotic stresses. Here, we examined the expression of *TgMATE* family members under various abiotic stressors using these transcriptomes of *Torreya grandis* under Al stress (50 µM AlCl_3_ treatment for 0, 8, and 24 h), high-temperature stress (40 °C high-temperature treatment for 0, 1, 2, and 3 days), low-temperature stress (2 °C low-temperature treatment for 0, 1, 2, 4, and 6 days), and drought stress (drought without watering for 0, 40, and 60 days). The Fragments per Kilobase Million (FPKM) data of *TgMATE* family genes from the transcriptomes were used to generate a gene expression matrix of *TgMATEs*. The up-regulated gene was defined as having a Fold Change value greater than 1.5, and the Fold Change of the down-regulated gene was less than 0.8. Finally, TBtools software v2.042 was utilized to visualize the gene expression heat map of *TgMATEs* [27].

### 4.6. Plant Materials and Al Stress Treatment

One-year-old *Torreya grandis* seedlings (bought from Zhuji City, China) were rinsed with sterile water and subsequently transferred to a container filled with 1/5 of Hoagland’s nutrients [66]. During the adaptation period, the seedlings were cultured for 15 days, and Hoagland’s solution was changed every 3 days. After that, they were moved to a 1/30 Hoagland nutritional solution [66] with 50 μM AlCl_3_ and 1 mM CaCl_2_ (pH 5.0). Root tips of 0–1 cm length of *Torreya grandis* seedlings were taken after treatment at 0, 8, and 24 h. For each treatment, three biological replicates were carried out.

### 4.7. RNA Extraction and qRT-PCR Validation

An RNAprep Pure Plant Plus Kit (Tiangen Biotech, Beijing, China) was used to extract total plant RNA from *Torreya grandis* seedling samples, which was then reverse transcribed into cDNA via a FastQuant RT Kit (with gDNase) (Tiangen Biotech, Beijing, China). Based on the findings of phylogenetic analysis, 7 *TgMATE* genes in clade Ⅳ that may have the ability to resist Al toxicity were chosen for qRT-PCR validation. The fluorescent quantitative primers used for the qRT-PCR are detailed in Appendix A, and the reaction steps were performed according to the method of Luo et al. [67]. There are three technical duplications that were conducted for each sample, and the relative expressions were calculated according to a 2^−ΔΔCt^ algorithm [68].

### 4.8. Weighted Gene Co-Expression Network Analysis

Transcription factors in *Torreya grandis* were forecasted through the PlantTFD (http://planttfdb.gao-lab.org/, accessed on 26 June 2023) [69]. To establish the gene co-expression networks, the WGCNA shiny program in TBtools was used [27]. The gene set is first filtered to ensure that every gene participating in the WGCNA has an expression level greater than 1 in at least 90% of the samples. A soft threshold power value of 14 was chosen to build a network. The minimum module size was chosen as 50, and the threshold for similar module merging was set to 0.35. Finally, the Cytoscape software v3.7.2 was used for the visualization of the interested module [70].

### 4.9. Dual-Luciferase Reporter Assay

A transient dual-luciferase experiment was performed to examine the transactivation activity of the co-expression transcription factors in the promoters of *TgMATE2*. Whole coding sequences of transcription factors were cloned and placed in the pGreenII 62-SK plasmid as the effector, while the promoter region of *TgMATE2* was placed in the pGreenII 0800-LUC plasmid as the reporter. Primers used in the PCR process are detailed in Appendix A. The effector and reporter were then converted into the *Agrobacterium tumefaciens* receptive bacteria strain GV3101 (with psoup) (Weidi, Shanghai, China) by the chemical transformation methods. After overnight incubation (28°, 200 rpm) in Luria–Bertani medium containing kanamycin and rifampicin, the target strains were collected and then re-suspended with infection buffers (10 mM morph ethanesulfonic acid, 10 mM MgCl_2_, 50 mM acetyl syringone, and pH 5.6) and the optical density was adjusted to 0.8. Consequently, the effector and the reporter were injected into the 5-week-old tobacco leaves at a ratio of 10 to 1. After 2 days of culture in the artificial climate chamber, intensities of firefly luciferase (LUC) and renila luciferase (Rluc) were detected according to the methods as demonstrated before [53]. A fully automated Tanon 5200 chemiluminescent image analyzer was used to observe fluorescence.

## 5. Conclusions

In this study, a comprehensive analysis of the *MATE* gene family in *Torreya grandis* was conducted. A total of ninety *MATE* genes were identified in the *Torreya grandis* genome, which can be phylogenetically classified into five clades. The conserved motifs and gene structure analysis revealed significant conservation of *TgMATEs* within each clade, which confirmed our classification scheme. The tandem duplication plays an important role in the expansion of the *TgMATE* gene family. Promoter analysis revealed that the majority of *TgMATEs* contain hormone-response and environmental stress-related elements. The expression profiles of *MATE* family members in *Torreya grandis* under Al, drought, high-temperature, and low-temperature stress confirmed that *TgMATEs* played critical roles in response to abiotic stress. Furthermore, potential transcription factors interacting with *TgMATEs* were identified by WGCNA analysis, and the dual-luciferase assay validated their credibility. Taken together, these findings provide a comprehensive and systematic characterization of the *MATE* gene family in *Torreya grandis* and provide a molecular basis for the stress-tolerant breeding of *Torreya grandis*.

## Figures and Tables

**Figure 1 ijms-25-03859-f001:**
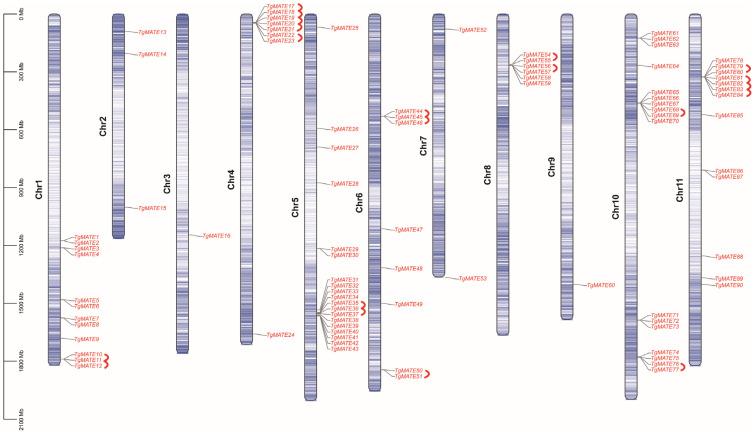
The distribution of the *TgMATE* gene family on the chromosomes and the occurrences of tandem duplication events (presented in red brackets).

**Figure 2 ijms-25-03859-f002:**
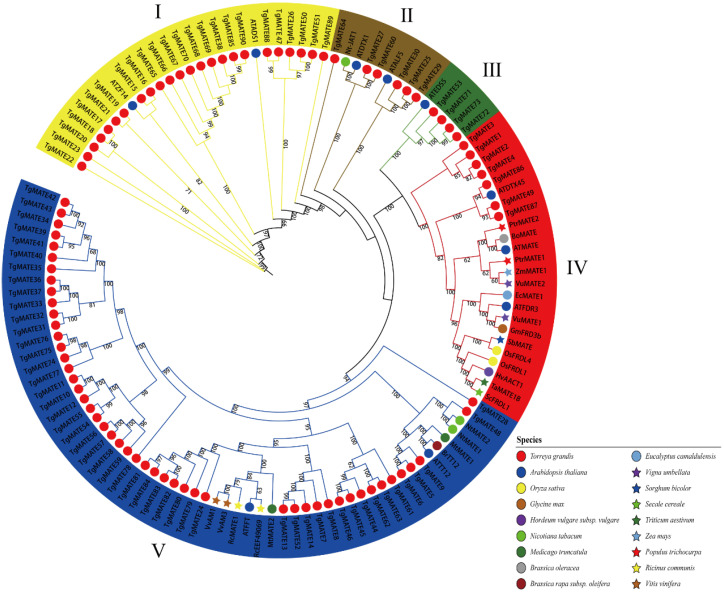
The phylogenetic tree of the expansion MATE genes in *Oryza sativa*, *Nicotiana tabacum*, *Arabidopsis thaliana*, *Ricinus communis*, *Brassica oleracea*, *Brassica rapa subsp. Oleifera*, *Eucalyptus camaldulensis*, *Glycine max*, *Hordeum vulgare subsp. Vulgare*, *Medicago truncatula*, *Populus trichocarpa*, *Secale cereale*, *Sorghum bicolor*, *Torreya grandis*, *Triticum aestivum*, *Vigna umbellate*, *Zea mays,* and *Vitis vinifera*. The Roman numerals I through V represent the five evolutionary groups of the MATE family.

**Figure 3 ijms-25-03859-f003:**
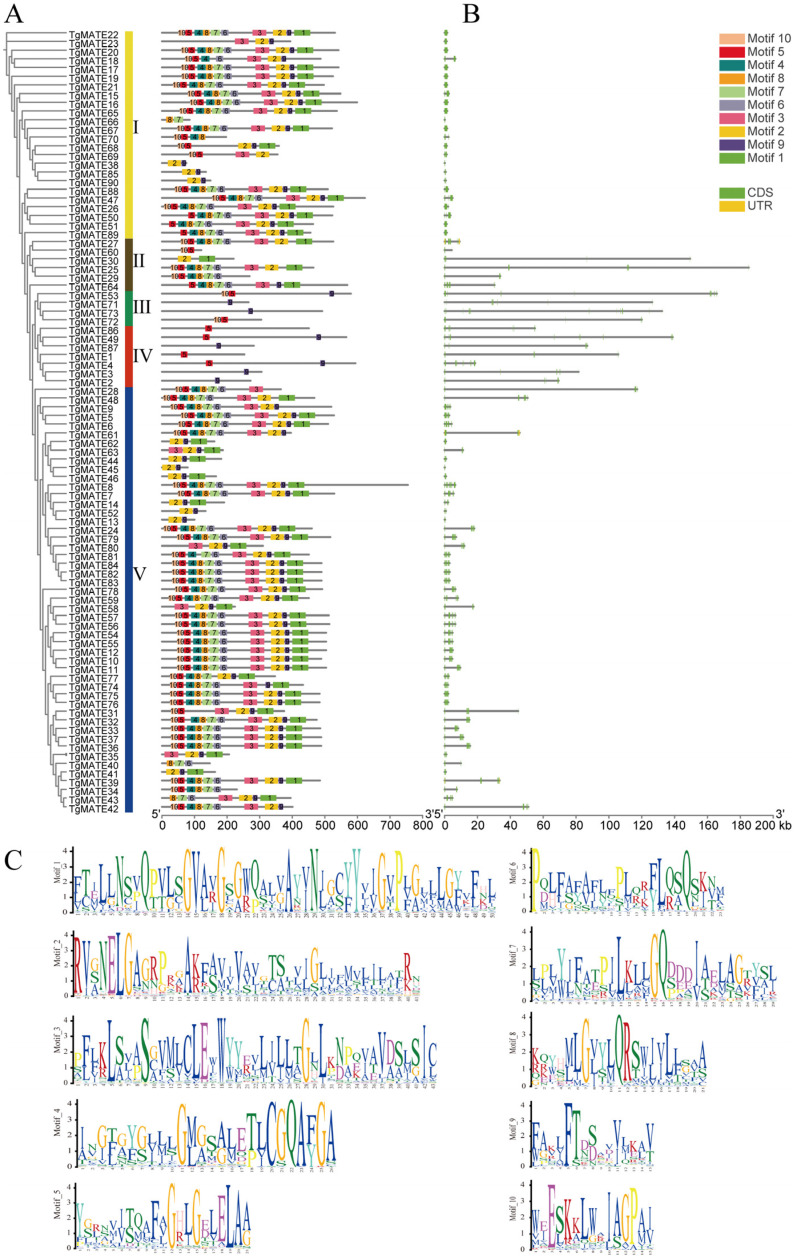
Protein motifs and gene structure of *TgMATEs*. (**A**) The conserved motif information of the TgMATE proteins. The Maximin Likelihood approach was used to build the phylogenetic tree of the TgMATE proteins. (**B**) Gene structure of the *TgMATE* genes. Yellow boxes are used to symbolize the untranslated regions (UTRs), and the green boxes represent the CDS areas. The introns are represented by the scale bar. (**C**) Conserved motifs of TgMATE proteins. The different font colors represent different amino acids.

**Figure 4 ijms-25-03859-f004:**
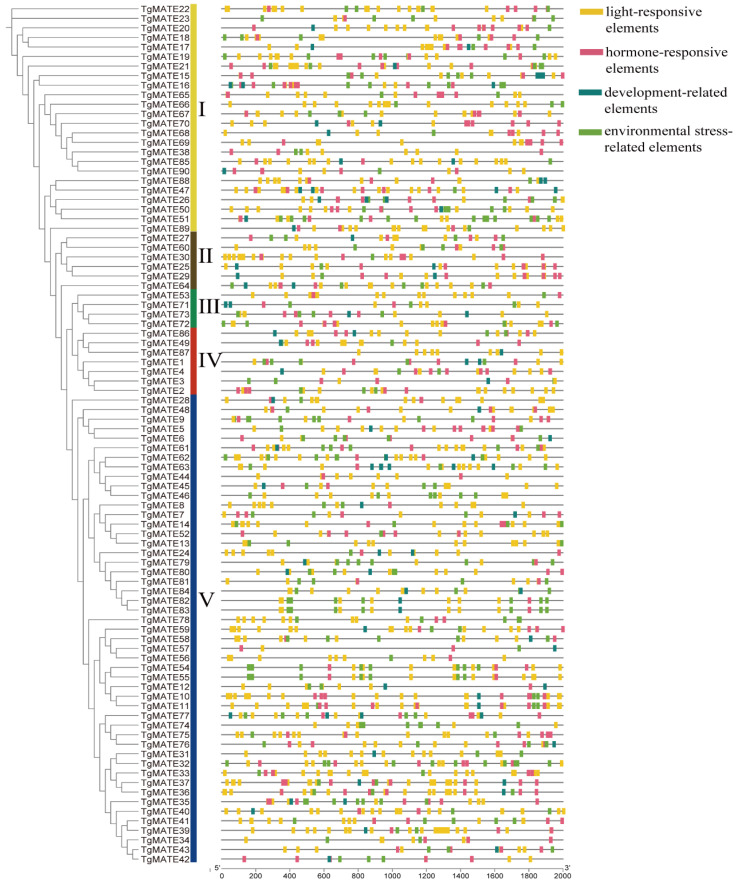
The different categories of cis-acting elements in *TgMATE* gene promoters. The Roman numerals I through V represent the five evolutionary groups of the TgMATE family.

**Figure 5 ijms-25-03859-f005:**
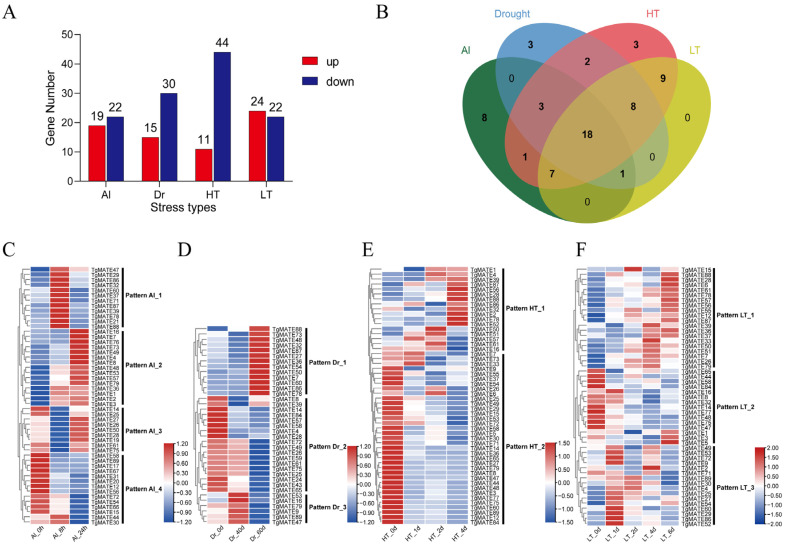
The expression patterns of the *TgMATE* family genes in response to aluminum (Al) stress, drought (Dr) stress, high-temperature (HT) stress, and low-temperature (LT) stress in *Torreya grandis*. (**A**) The numbers of up-regulated and down-regulated *TgMATE* genes under Al, Dr, HT, and LT stress. (**B**) Venn diagrams show the overlap of differentially expressed *TgMATEs* in response to Al (green), Dr (blue), HT (red), and LT (yellow). (**C**–**F**) Heat maps show the expression patterns of the stress-responsive *TgMATEs* under Al (**C**), Dr (**D**), HT (**E**), and LT stress (**F**). The FPKM values are log-normalized and clustered in rows.

**Figure 6 ijms-25-03859-f006:**
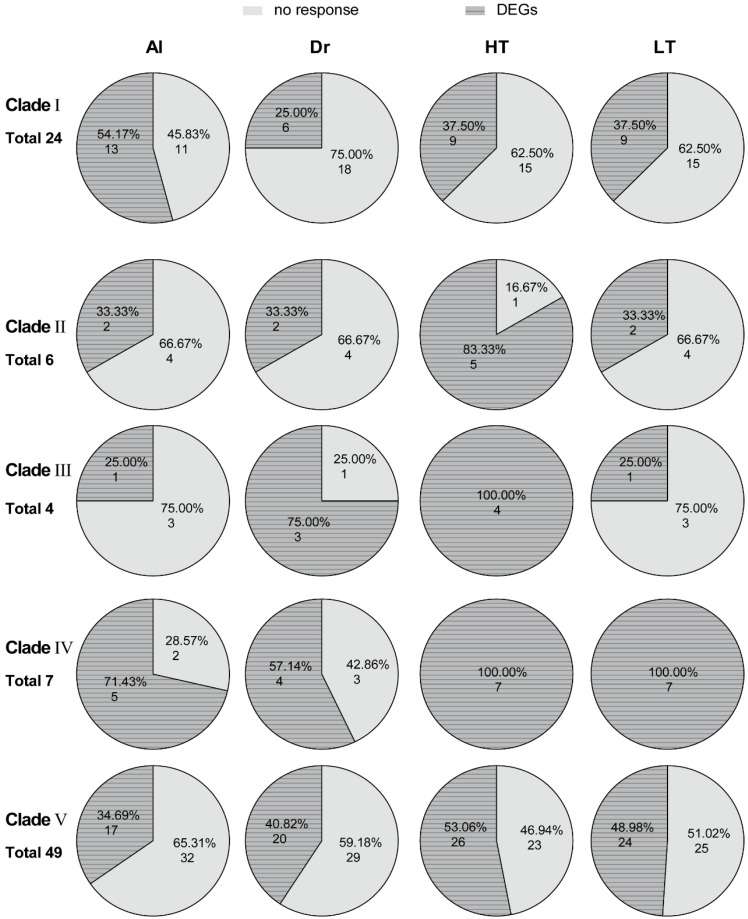
The proportion of differentially expressed genes (DEGs) and unresponsive genes in different polygenetic clades of *TgMATEs* under Al stress (Al), drought stress (Dr), high-temperature stress (HT), and low-temperature stress (LT).

**Figure 7 ijms-25-03859-f007:**
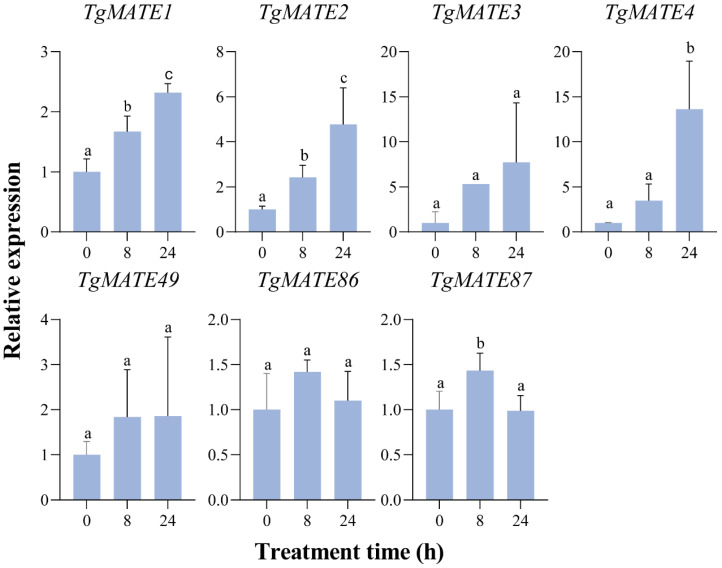
qRT-PCR analysis of the 7 *TgMATE* genes from clade IV under Al stress. The gene expression is normalized relative to the *Torreya grandis* reference gene, *TgActin*. The data present the average of three biological replicates, and ANOVA was employed to test the hypothesis. Different letters represent statistically significant *p*-values that are less than 0.05.

**Figure 8 ijms-25-03859-f008:**
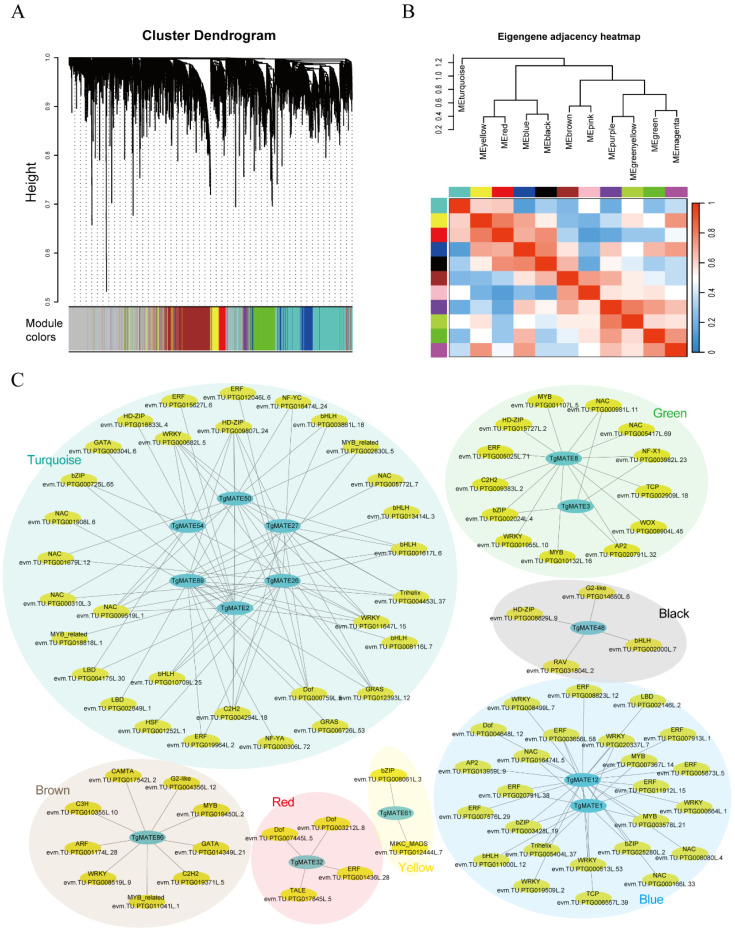
Relationships between *TgMATEs* and possible transcription factors predicted by WGCNA methods. (**A**) Determination of soft threshold power. (**B**) Eigengene adjacency heatmap of WGCNA. (**C**) Co-expression network represents the relationships between *TgMATEs* and possible transcription factors predicted by WGCNA methods. The blue ovals represent the *TgMATE* genes, and the yellow ovals represent transcription factors. Different background colors represent different modules.

**Figure 9 ijms-25-03859-f009:**
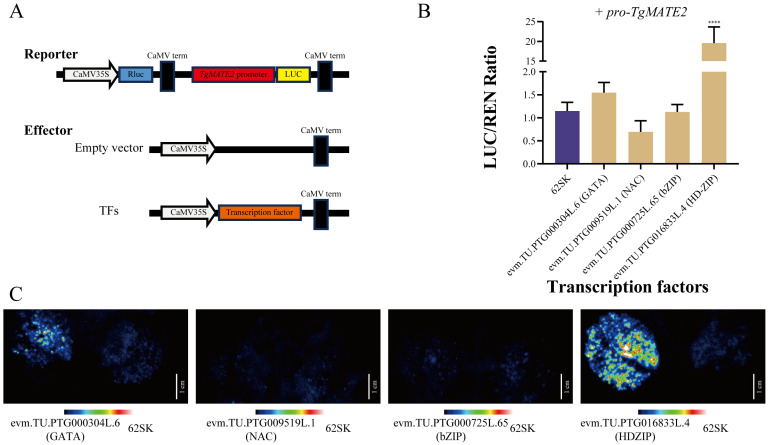
Dual-luciferase experiment between *TgMATE2* and its candidate transcription factors. (**A**) Structure schematic of reporter and effector. (**B**) Transcriptional activation ability of 4 transcription factors for *TgMATE2* in tobacco leaves. Each data presents the average of three biological replicates. A quadruple asterisk denotes the significant difference (*p* < 0.0001) assessed by ANOVA. (**C**) Fluorescence is released as a result of the reaction between the substrate and firefly luciferase. The scale bar corresponds to 1 cm.

## Data Availability

The raw data supporting the conclusions of this article will be made available by the authors on request.

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
