# Peer review of "Genome-Wide Identification, Expression Analysis under Abiotic Stress and Co-Expression Analysis of MATE Gene Family in Torreya grandis"

_ijms, 2024, doi:10.3390/ijms25073859_

Round 1

Reviewer 1 Report

Comments and Suggestions for Authors

The research systematically analyzed the genome of Torreya grandis and discovered 90 TgMATE genes, which were classified into 5 clades based on phylogenetic relationships and gene structures. Analysis of cis-acting elements in gene promoters revealed their association with environmental stress. Transcript profiling showed differential expression of 63 TgMATE genes under various stresses, indicating their role in stress response. Additionally, WGCNA analysis identified interactions with transcription factors, validated by a Double-luciferase assay. However, according to existing literature, many studies have already demonstrated that the results of similarity analysis provide comprehensive information on the MATE gene family and aid in understanding the functional mechanisms of MATE genes. Although the analyzed subjects differ, the studies already published have all focused on the MATE gene family, and the methods, figures, and results employed in these papers are highly similar to those in our study. So, what is the innovation point of this article? Therefore, the article had major drawbacks either in study design or the innovation. In my opinion, it is not acceptable for publication in its present form.

1.       Santos A L, Chaves-Silva S, Yang L, et al. Global analysis of the MATE gene family of metabolite transporters in tomato[J]. BMC plant biology, 2017, 17(1): 1-13.

2.       Li Y, He H, He L F. Genome-wide analysis of the MATE gene family in potato[J]. Molecular biology reports, 2019, 46: 403-414.

3.       Manzoor M A, Li G, Abdullah M, et al. Genome‐wide investigation and comparative analysis of MATE gene family in Rosaceae species and their regulatory role in abiotic stress responses in Chinese pear (Pyrus bretschneideri)[J]. Physiologia Plantarum, 2021, 173(3): 1163-1178.

And so on….

Reviewer 2 Report

Comments and Suggestions for Authors

Journal IJMS (ISSN 1422-0067)

Manuscript ID ijms-2881098

Type Article

Title Genome-wide identification, expression analysis under abiotic stress, and co-expression analysis of MATE Gene Family in Torreya grandis

Authors Hang Shen , Ying Hou , Xiaorong Wang , Yaru Li , Jiasheng Wu * , Heqiang Lou *

Section Molecular Plant Sciences

Special Issue Advances in Genetics and Phylogenomics of Tree

Dear Editor,

Thank you for taking the manuscript seriously for peer review.

The publication outlines important aspects that have application relevance.

However, the text needs some additions and corrections. I will post below my comments on the text.

Sincerely

Reviewer

Keywords

1 Eliminate words that appear in the title

Abstract

2.Strengthen the structure of the abstract by retaining:

Introduction

l. 41-45.

3 Please elaborate on the thought, give details, examples of "ensuring food security".

"... consequently, the breeding of Torreya grandis with greater abiotic stress tolerance is critical for ensuring food security".

4. I suggest completing the use value of this species with TORREYA GRANDIS SEED OIL, among others.

l. 46-57

5. Replace background information with recent substantive reports closely related to the topic of the paper.

l. 59-74

6. Please list specific metabolites (there are many), state in which species and in which organ (raw material) - strictly referring to the topic of the manuscript.

7. Explain what they consist of, the various activities "...MATE transporters are engaged in a variety of biological activities,...".

8. Explain what hormone signaling "...hormone signaling...".

l.74-86

9. Please formulate the scientific research thesis

10. Please reinforce the rationale for addressing the research topic

11. Formulate the specific purpose of the work, and transfer the methodological activities to the subsection material and methods of research.

l. 80 - 86

12. This information not in this section.

Results

13. figure 2. we write Latin names diagonally.

14. Figure 5A, 7, pB. Insert X-axis designation.

Discussion

l. 310 - 320

15. Please complete (if there is data) the comparison also in the nematode plants.

l.321-330

16.Discussion is not the place to quote figures.

17. Please discuss the results obtained with the achievements of other authors (no discussion in the subject paragraph).

l. 331-361

18 Please, see comment number 15.

l. 362-381

19. please, see comment number 15.

l.382-410

20. please, see comment number 15.

21.Indicate the application use of the obtained results of the study.

Methods

l. 456-463

22. Complete the literature citation on Hoagland's nutrients

l. 464-472.

23. Complete the literature citation confirming the correctness of the methodological measures

Conclusion

24. note whether the conclusions respond to the scientific research theses (when completed)

References

25. please check the cited literature step by step.

26. Please make corrections in accordance with the instructions for authors.

l. 539-544.

27.Pay attention to the notation of titles.

l. 559-561.

28.The names of genus and species are written obliquely, etc. It is impossible to list them all

Reviewer 3 Report

Comments and Suggestions for Authors

The contents were worth to study and some useful conclusions were obtained. Reviewer recommends the publication of this paper in this journal. However, before acceptance, the following major concerns must be addressed:

1. Please reorganize the language of the Abstract, the first half is illogical and the data displayed in the abstract should be revised.

2. The language and the grammar of the article need to be optimized.

3. The format of the paper should be united

4. Try to avoid excessive use of abbreviation in this section. In fact, many abbreviations are used in the main text, which may hamper the readability of this manuscript.

5. How many times have the tests been replicated? Error bars all through the data points are narrow and likely abnormal. I doubt the accuracy of the results. Please check this. Fig. 7

6. Compare and discuss more about abiotic stress related genes in reference genome. Lack of discussion and mechanism. 

7. "The collinearity analysis of the Torreya grandis genome revealed there are many tandem repeat genes in the TgMATEs (Fig. 1), suggesting the gene tandem duplication has encouraged  the expansion of the MATE family in Torreya grandis."  Please discuss this mechanism rather than observing. 

Round 2

Reviewer 3 Report

Comments and Suggestions for Authors

The concerns raised during the review process appear to have been
addressed comprehensively. The manuscript now meets the standards for
publication in the journal.